# Hollow Fiber Polyimide Membranes Prepared in a Triple Orifice Spinneret: Effect of a Reduced Water Activity in the Bore Fluid on the Gas Separation Performance

**DOI:** 10.3390/polym13132211

**Published:** 2021-07-04

**Authors:** Paola Bernardo, Franco Tasselli, Gabriele Clarizia

**Affiliations:** Institute on Membrane Technology, ITM-CNR, Via P. Bucci 17/C, 87036 Rende, Italy; p.bernardo@itm.cnr.it (P.B.); f.tasselli@itm.cnr.it (F.T.)

**Keywords:** hollow fiber, polyimide, gas separation, bore fluid water activity

## Abstract

Polyimide-based hollow fibers were spun using a triple orifice spinneret in order to apply them in gas separation. The membrane structure was tailored producing a porous external layer and a thin internal skin layer, that controlled the gas transport. The measurement of gas permeation rates and the morphological analysis were combined to obtain information on the performance of the membranes. The aim was to tune the inner top layer and investigate the role of the bore fluid on the gas permeation properties of the membranes. The bore fluid composition was explored by using water mixtures containing the solvent used for preparing the dope solution or a salt in order to reduce the water activity in the inner coagulant, but also a low amount of a crosslinker for improving the gas selectivity. The change of the dope flow-rate was also analyzed. At moderate dope flow-rates, the use of a saline water solution as bore fluid is more effective in enhancing the membrane gas selectivity with respect to a bore fluid containing certain amounts of solvent. This option represents a green approach for the preparation of the membrane. The behavior of the prepared hollow fibers over time (physical aging) in gas permeation was discussed.

## 1. Introduction

Hollow fibers (HFs) provide a high packing density, resulting in extremely compact modules that treat huge feed streams in reduced spaces, thus represent the dominant membrane configuration in the case of gas separation [1]. The presence of a dense top layer is a prerequisite to perform the separation of small molecules such as permanent gases, thus sophisticated asymmetric membranes are required for this application. Apart from skin thickness, the morphology across the entire membrane thickness also influences its ultimate performance. The HF morphology can be modulated by playing on different parameters during the spinning process. Typically, an increase of polymer concentration in the dope solution results in a denser structure with thicker skin layers [2]. The addition of a non-solvent or a worse solvent to the dope brings the composition closer to the binodal line in a ternary diagram, accelerating the phase separation [3]. Water is largely used as the non-solvent (NS), being the most environmentally friendly fluid and for the good compatibility with common organic solvents. It has a strong coagulation power. The presence of water both in the bore fluid (BF) and in the coagulation bath causes polymer precipitation with the formation of a double skin localized on the internal and external surfaces of the HFs.

The spinneret design provides other options to tailor the HF morphology. Typically, it is a double orifice die in which the BF flows co-axially with the dope solution, generating the hollow fiber when the extruded material enters in contact with the coagulation medium. Instead, in a triple orifice spinneret, a bore liquid (inner), a polymer solution (intermediate), and an additional fluid (external) are extruded at the same time. This setup allows either the co-extrusion of two different polymeric dopes (dual-layer membranes) [4] or the engineering of the outermost layer of the HFs by using an external fluid based on a solvent for the polymer [5,6,7].

In previous works, we proved it was possible to tailor the morphology and performance of polymeric HFs for gas separation by using a triple orifice spinneret for the dry-jet/wet spinning and by selecting specific post-treatment protocols [7,8,9]. In particular, by working on the outer layer in order to create a porous surface, it is possible to enhance both the gas permeance and selectivity of the HF membranes intended for gas separation [7]. Indeed, the inner skin layer becomes the main resistance that controls the gas transport.

Once the transport resistance is limited to bore side of the membrane, it is possible to modify the inner separating layer by an appropriate formulation of the BF composition. Indeed, the wet-phase inversion process involves an exchange between the solvent from the nascent HF and the NS from the coagulant driven by concentration gradients. The kinetics of the process are affected by different parameters. The addition of an additive to the internal coagulant, through the lowering of the water activity, results in a reduction of the chemical potential difference between the solvent and non-solvent and, thus, slows down the phase inversion process (liquid–liquid demixing). A delayed process can result in an integrally skinned asymmetric membrane, where a skin layer and microporous substructure are formed simultaneously during the phase inversion process.

In the case of polysulfone HF membranes developed for gas separation, a BF with lower water activity produced more selective samples [10,11]. Pesek and Koros prepared polysulfone HF membranes, achieving a better control of the two-phase separation processes that occur at the same time at the internal and external surfaces by lowering the water activity of the bore fluid [10]. The hypothesis is that bore fluids having a low water activity slow down the solvent diffusion into the bore and prevent the active layer from perforation. Ismail et al. obtained polysulfone hollow fibers displaying a high selectivity after coating with silicone by combining a low water activity of the bore fluid with a high extrusion rate [12].

The change from one polymer to another is not trivial, therefore, we analyzed the role of the bore fluid composition in a triple orifice spinneret setup on the performance of polyimide-based HFs. In particular, we used aqueous solutions containing certain amounts of the same solvent used for the dope preparation or a salt. Both additives reduced the coagulation strength of water, but the salt solution also promoted the solvent outflow from the nascent differently from the solvent solution. NaCl was chosen as an inexpensive and non-toxic alternative additive for the bore fluid. Greener preparation methods for the polymeric membranes are under investigation and mainly focus on the replacement of the commonly used solvents [13,14]. In particular, the selected concentration of NaCl was of 20 wt.%, as typically encountered in concentrated brine solutions available from desalination operations [15].

## 2. Experimental

### 2.1. Materials

The 3,3′,4,4′-benzophenonetetracarboxylic dianhydride and diaminophenylindane polyimide (Matrimid^®^5218) was supplied by Huntsman Advanced Materials (Everberg, Belgium). N-Methyl-2-pyrrolidone (NMP, purity of 99%) was purchased from VWR International (Milan, Italy) and used as a solvent for the preparation of the HFs due to its high boiling point and miscibility in water. Sodium chloride (NaCl, Sigma Aldrich (Milan, Italy) and ethylene diamine (EDA, Sigma Aldrich (Milan, Italy) were dissolved in de-ionized water and used as additives for the bore fluid. The chemicals were used as received.

A bi-component epoxy resin (Elan Tech^®^, supplied by ELANTAS Italia U.R.L., Ascoli Piceno, Italy) was used for sealing the prepared HFs in the modules. He, H_2_, N_2_, O_2_, CH_4_, and CO_2_, with a purity of 99.99% (SAPIO, Monza, Italy), were used as single gases for the permeation rate tests.

### 2.2. Spinning Protocol

The HF membranes were prepared according to a dry-jet, wet-quench spinning process. The original spinning pilot plant is described elsewhere [16]. The setup, modified to use a triple orifice spinneret, has an additional gear pump for co-feeding the external fluid. The spinneret has nominal dimensions of 1.3/0.8/0.4 mm for the outer, internal, and needle diameters, respectively.

Dope solutions were prepared by dissolving the polyimide in NMP. The polymer powder was slowly added to the solvent into a glass flask in order to prevent its agglomeration, up to a concentration of 24 wt.%. The solution was left overnight under mechanical stirring (ca. 300 rpm), keeping a temperature of 50 °C. The air bubbles, formed during the stirring, were removed from the dope by applying vacuum for few minutes. The dope was kept at 50 °C in a tank under an inert nitrogen atmosphere and all the lines from the dope tank to the spinneret are heated in order to ensure the fine rheological behavior of the concentrated dope solution. This operation mode prevented polymer precipitation before release from the spinneret. On the other hand, dope solutions having a low viscosity form discontinuous strands. Therefore, an appropriate balance between polymer concentration and temperature for the spinning was required to assure the right dope viscosity.

The composition of the external fluid (EF) was: NMP/water = 95/5 on a weight basis. A quite high air gap was fixed to allow a sufficient contact time between the nascent fiber and the EF before the coagulation bath. Tap water was used as the coagulation medium for the external HF surface. The bore fluid (BF) was alternatively pure water, a 20 wt.% aqueous solution of sodium chloride, or a 20, 30, 60 wt.% aqueous solution of NMP.

The operating conditions are listed in Table 1, while the composition of the BF and EF of the different spun batches are illustrated in Table 2.

The effect of the dope flow-rate was examined by using two values and keeping the same value for the BF and EF flow-rate.

### 2.3. SEM Analyses

The morphology of the prepared HFs was studied using an EVO|MA 10 (Zeiss, Italy) Scanning Electron Microscope (SEM) at an acceleration voltage of 20 kV. In order to examine the sample cross-section, avoiding their deformation, the fibers were fractured in liquid N_2_ and then coated with a gold layer using a sputter-coater.

### 2.4. Gas Permeation Tests

The fixed volume/variable pressure equipment used for the permeation tests is described elsewhere [17]. The pressure-normalized fluxes of the gases (permeance) were measured at 25 °C and at a pressure drop of 1 bar, keeping the permeate side under vacuum. Membrane selectivity was calculated as the ratio of the pressure-normalized fluxes of two pure gases. The permeation tests were carried out on the HFs directly dried from the water–wet state and after an aging period of 18 months.

## 3. Results and Discussion

### 3.1. Morphological Analysis

The morphology of the prepared HFs is clearly asymmetric, with a very thin dense skin on the inner side and a thick microporous sublayer.

Some representative samples, prepared at Q_DOPE_ = 3.6 g min^−1^, are shown in Figure 1 and Figure 2 (a cross-section and magnification of the inner skin layer). The other batches prepared at Q_DOPE_ = 5.0 g min^−1^, are shown in the Appendix A.

Figure 1 shows the cross-section of the HFs prepared using a BF containing water or NMP/water, 20/80; instead, Figure 2 reports the images of the HFs prepared using a BF containing EDA/water or NaCl/water, 20/80.

From these SEM images it can be seen that the presence of the solvent or the salt in the BF makes the skin layer thinner (see Table 3), creating a sublayer with smaller fingers and a thicker spongy layer, somewhat reducing the macrovoids propagation. The support offers a very low resistance to the gas transport, providing the mechanical strength, whereas the dense top layer accomplishes the separation role. This suggests that only the very thin layer controls the separation performance achieving the productivity enhancement for the system. However, as discussed in Section 3.2, the substructure underneath the top layer also has a role in determining the final performance of the HFs for gas transport.

### 3.2. Gas Permeation

The gas permeation tests provided experimental evidence quantifying the effects of the preparation conditions on the membrane structure discussed in the morphological investigation.

Table 4 and Table 5 report the main gas permeation properties of the HFs prepared at a dope flow-rate of 5.0 g min^−1^ and 3.6 g min^−1^, respectively, tested as prepared. The data reported in Table 4 for the batch H4 are taken from our previous work [7].

The HFs were defect-free, displaying good selectivity for different gas pairs. Therefore, the membranes did not necessitate silicone coating, a standard protocol applied to heal eventual defects in the selective layer [18].

According to Pesek and Koros [19] if the selectivity of a HF overcomes 80% of the intrinsic selectivity of a dense film them the gas transport is mostly regulated by the solution/diffusion mechanism. Comparing our results with those measured on a homogeneous flat membrane based on the same polymer/solvent combination (Matrimid^®^/NMP) [7], we observed that this condition occurs whenever an enriched-solvent external fluid is used for HF spinning. Indeed, the use of a solvent-rich EF lead to HFs with a porous external surface, and therefore, the inner dense layer is responsible for the main gas transport resistance [7]. The HF configuration in which a dense skin layer is localized only on the inner side is particularly suited for gas separation since it improves both gas permeance and selectivity [7]. In addition, protection issues for the thin selective layer of the HFs favor this membrane design. The counter-current flow pattern for feed and permeate, that maximizes the extent of separation, is best achieved with tube-side feed. This operation mode ensures a monodirectional flow on the feed side and permeate [20]. This is a further reason for the adoption of a solvent-rich EF in the HF preparation.

In the case of asymmetric structures, the apparent gas transport properties are due to the skin layer but also to the underlying porous substructure [21].

The contribution of the substructure was estimated by considering the asymmetric HFs as a series of resistance to the gas transport. In this evaluation, we assumed the selectivity for the active layer coincident with the intrinsic selectivity of the material (measured on dense film in our previous paper [7]) and a Knudsen mechanism for the porous substructure. Thus, on the basis of the measured selectivity for the asymmetric HFs, we evaluated the relative contribution of the substructure to the total resistance. We referred to small/large gas pairs (e.g., H_2_/N_2_) using the following equation:$$\frac{R_{skin}}{R_{TOT}}\left(\%\right)=\frac{\alpha_{skin}-\alpha_{HF}}{\alpha_{substructure}-\alpha_{skin}}*100$$

This ratio is equal to 7.87% for H0, 2.57% for H1, 3.29% for H5, 12.0% for H3, and 14.9% for H4. In the other cases (H2, H6, H7), the selectivity of the HF batches, being higher than the intrinsic value measured for an isotropic thick film, suggests a densification of the active layer and thus good permselective properties. Densification is typically observed for thin layers of glassy polymers [22]. At the same time, a better selectivity is expected for the crosslinked inner layer in the H8 membranes.

#### 3.2.1. Effect of the Dope Flow-Rate

Two flow-rate values were used for the dope extrusion. Some HF batches were spun at 5.0 g min^−1^ to make a comparison with previous works [7,9], while a lower flow-rate (3.6 g min^−1^) was adopted in order to increase the contact time of the nascent HF with the EF. A reduced take-up speed allows the nascent fiber to go through the air gap for a longer time period. This condition also affects the coagulation process on the inner region. Indeed, the reduced dope flow-rate allows a better exchange due to a longer contact time between the solutions on the bore side, resulting in the successful formation of the defect-free skin layer.

The HFs prepared with pure water as BF and an EF (NMP/water, 95/5) presented a good selectivity at both dope flow-rates, close to the intrinsic values typically reported for Matrimid dense films. In particular, the higher dope flow-rate lead to slightly larger gas permeance and selectivity. This effect can be related to an enhanced molecular orientation in the fiber skin obtained at higher extrusion rates, as also observed for polysulfone HFs [23].

It is worth noting that the manipulation of the dope flow-rate has some limits. Ismail et al. evidenced the risk of a dramatic loss of the membrane selectivity for samples prepared at dope extrusion rates above a critical point, resulting in relatively porous skin structures [23]. In our case, the permeation tests confirmed that this limit condition for the dope extrusion rate was not reached.

#### 3.2.2. Effect of the Bore Fluid Composition

In order to reduce the activity of the aqueous inner coagulant, different HF batches were spun by adding weighted amounts of the solvent to the BF, up to 60 wt.%. More concentrated NMP solutions did not allow a uniform spinning line in our setup and they were not further investigated.

The Hansen solubility parameters (HSPs) are useful to characterize the interaction forces between different materials [24]. The closer the HSP values, the greater the affinity between species. Indeed, the total solubility parameter (*δ*_t_) is derived from the cohesive energy density (CED) of the material [25]:$$\delta_{t}=\sqrt{CED}$$

The CED is obtained from the vaporization enthalpy and is also linked to the process of dissolution (i.e., the energy required to create cavities in a liquid in order to accommodate solute molecules) [26]. High CED values characterize highly polar solvents and/or reflect the presence of hydrogen bonds. Indeed, the total solubility parameter can be decoupled into three components: nonpolar, atomic (δ_D_) interactions; permanent dipole-dipole molecular interactions (δ_P_); and hydrogen bonding interactions (δ_H_) [24]:*δ*_t_^2^ = *δ*_D_^2^ + *δ*_P_^2^ + *δ*_H_^2^

Table 6 reports the HSPs and Gordon parameter of the polymer, solvent, and coagulants used in this work. The reported HSP data for the polymer, NMP, and water are taken from the literature. A weighted average was calculated for the NMP/water solutions. In the case of the salt solution, no data were available. Therefore, we evaluated the cohesive energy density for the NaCl crystals using data (7.9 eV/molecule unit) reported in [27]. However, as a further description of the material cohesiveness, the Gordon parameter (GP) was considered since it is suited to describing non-volatile species. It is defined as:$$GP=\frac{\gamma }{V_{m}{}^{1/3}}$$
where *γ* is the surface tension of the material and *V_m_* is its molar volume [28].

In the case of Matrimid, GP was determined by calculating the surface tension of the polymer from its water contact angle [29]:$$\gamma_{S}≅\gamma_{L}\frac{{\left(1+\cos\vartheta \right)}^{2}}{4\;F^{2}}$$
$$F=\frac{{\left(V_{S}\;V_{L}\right)}^{1/3}}{{\left(V_{S}{}^{1/3}+V_{L}{}^{1/3}\;\right)}^{2}}$$
where *γ_S_* is the surface tension of the solid (polymer), *γ_L_* is the surface tension of the liquid (water), *θ* is the water contact angle, and *V_S_* and *V_L_* are the molar volumes of the polymer and of water, respectively. A value of 75° was assumed for the water contact angle of the polyimide [30].

**Table 6 polymers-13-02211-t006:** Hansen solubility parameters (HSPs) and Gordon parameter (GP) of the polymer, solvent, and coagulants used.

Material	HSP (MPa)^0.5^				GP (J/m^3^)
	*δ* _D_	*δ* _P_	*δ* _H_	*δ* _t_	
Matrimid^®^	18.7	9.5	6.7	22.0 [31]	1.5 ^a^
NMP	18.0	12.3	7.2	22.96 [32]	0.89 [32]
Water	15.5	16.0	42.3	47.8 [32]	2.77 [32]
NaCl				168.0 ^b^	
NMP/water (20/80 wt/wt)	16.0	15.3	35.3	42.8 ^c^	2.39 ^c^
NMP/water (30/70 wt/wt)	16.2	14.9	31.8	40.3	
NMP/water (60/40 wt/wt)	17.0	13.8	21.2	32.9	
NMP/water (95/5 wt/wt)	17.9	12.5	9.0	24.2	
NaCl/water (20/80 wt/wt)				71.84 ^c^	4.32 ^d^

^a^ calculated using a solid surface tension evaluated from contact angle; ^b^ calculated as *CED*^0.5^; ^c^ calculated as the weighted average of the individual components; ^d^ calculated using the data (*γ* and *V_m_*) reported in [33] for a solution of NaCl/water 4.6 m (20.64 wt.%).

In analyzing the difference in the parameters reported in Table 6, the same trend is evident for the total solubility parameter and the GP. Indeed, some authors report a linear correlation between them for molecular liquids and for ionic liquids [34].

Water, characterized by a large hydrogen bond term, is a high strength non-solvent for Matrimid^®^, while NMP is a good solvent for the polymer as evidenced by their similar solubility parameters. Therefore, adding NMP to water reduces the difference of the solubility parameters between the polymer and the mixture of coagulant (Table 6), particularly when higher amounts of solvent are added to the coagulation medium.

In more detail, among the HFs prepared at Q_DOPE_ = 5 g min^−1^, the use of a low NMP concentration (20 wt.%) in the BF produced lower pressure-normalized fluxes than those obtained using pure water as the BF. This suggests that a reduced water activity of the BF tends to produce thicker effective skins, as also observed by Ismail et al. [12]. This behavior is associated to an important increase of selectivity for H_2_/N_2_ and a modest decrease for O_2_/N_2_ and CO_2_/N_2_ gas pairs compared to the samples spun using pure water as the BF.

On the other hand, a further reduced water activity, obtained by increasing the NMP concentration in the BF up to 60 wt.%, resulted in HFs with progressively larger gas permeance and lower selectivity.

The affinity for the solvent/coagulant pair is improved by the addition of NMP to the BF (Table 6), facilitating their mutual exchange. However, in this complex system, different variables play a role. Since the BF additive is the same solvent present in the dope solution, its addition to the coagulant reduces the concentration gradient that drives its out-flow from the nascent HF, and at the same time, by reducing the water activity in the BF, slows down the coagulant inflow. Therefore, a delayed demixing process could be expected.

In the presence of large amounts of solvent in the BF, the polymer chains remain flexible for a longer time. In addition, the viscosity of the coagulant plays an important role during the phase inversion process since it influences the membrane formation kinetics. Increasing amounts of NMP to water result in a mixture with an increased viscosity that passes through a maximum at a mole fraction of ca. 0.3 (corresponding to an NMP weight concentration of 70%) [35]. This is due to deviation from ideality for the strong interactions between NMP and water [36]. In the investigated range of NMP concentration, the viscosity of the solution increases with NMP concentration: the BF containing 20 wt.% of NMP has a viscosity of ca. 2 cPoise, while at 60 wt.% of NMP the viscosity is close to the maximum for this mixture (ca. 5 cPoise at 25 °C). Therefore, the larger viscosity of the more concentrated BF hinders solvent-exchange progression and the process is confined to a thinner outer region. In addition, polymer redissolution at higher NMP concentrations in the BF may be responsible for the creation of the thinner skin layer with some “defects” and a consequent more relevant role of the substructure to the global transport. As a result, the permselectivity of the HFs is reduced.

The lowest NMP concentration in the BF used in this work (20 wt.%) lead to better selective properties for small/large gas pairs (e.g., H_2_/N_2_) due to a molecular sieving transport mechanism. Therefore, this BF composition was adopted also at a reduced dope flow-rate. The HFs spun with 20% NMP-water solutions as BF had no significant variations on the permeation parameters when changing the dope flow-rate, as reported in Table 4. Accordingly, the beneficial effect of the NMP addition at 20% to the BF on the selectivity was confirmed compared to pure water.

In order to further reduce the activity of the aqueous inner coagulant, without compromising the HF selectivity as observed at increasing NMP concentrations, other HF batches were spun using NaCl as additive for the BF. Indeed, owing to the electrolytic nature of the salt, its addition was more effective in reducing water activity [37]. The concentration of the salt solution (20 wt.%) is close to its saturation limit.

The NaCl addition to the BF increases the pressure-normalized flux of the HFs for all tested gases and the selectivity values with respect to those HFs prepared using only water as bore fluid. The larger gas permeance suggests a thin effective skin layer on the internal side of the prepared HFs.

In this case, the presence of a salt was more effective than the same weight amount of solvent in the BF, improving either the gas permeance or the CO_2_/N_2_ selectivity. This is a result of the combined effect of the viscosity and water activity of the BF. The salt solution at 20 wt.% has a slightly lower viscosity (1.5 cPoise) than the same weight in water of NMP (2 cPoise). In addition, the solvent has a larger molecular size, hindering the its diffusion. Contrary to NMP, the addition of salt to water increases the difference between the polymer and coagulation fluid (see the GP reported in Table 6), also with respect to the water alone, making them even less compatible. This implies a faster polymer precipitation. On the other hand, different from the solvent, the salt in the BF may induce an entanglement of the polymer chains, thus moving the composition close to precipitation point and resulting in a quicker phase separation [38].

In terms of comparison, other HFs were spun at a lower dope flow-rate with a BF containing a crosslinker agent (EDA) for the polyimide (H8) displaying a distinct size-sieving behavior, as indicated by their greater H_2_/N_2_ selectivity. The tightening effect induced by the crosslinker on the polymer matrix provided selectivity values larger than the intrinsic values typically reported for Matrimid dense films, at the expense of a lower permeance.

Interestingly, as proof of the effectiveness of the adopted spinning protocol, the HFs prepared with the salt solution as BF achieved selectivity values comparable with those obtained by using a diluted quantity of crosslinker in the BF, however, they were associated with higher gas permeance (at least three times higher). The highly engineered active layer obtained in these conditions overcame the selectivity values intrinsic of this material.

#### 3.2.3. Long Term Performance

In order to evaluate their durability, the HFs were tested over time, after 18 months from their spinning. As commonly observed for membranes based on polyimides [9,39], the HFs displayed a reduced gas permeance, but the selectivity increased upon aging (Table 7). However, some specific peculiarities occurred for the different samples.

First of all, all samples maintain their integrity, assuring appealing performance over time. The HFs prepared without the EF were less resistant to physical aging, showing the more relevant permeance decay and even keeping their permselective properties. In particular, the permeance of the smaller gases (H_2_ and He) halves, whereas that of the less permeable components (CH_4_ and N_2_) is reduced to one third.

The presence of the EF improved the transport properties of the “as-spun” samples, spun at both dope flow-rates. In fact, for all gas species the reduction in their permeance (H1 and H5 samples) was lower than in the H0 samples, and the results of the permselectivity for the gas pairs was always higher. Furthermore, the reduction of water activity in the BF due to addition of NMP (20 wt.%) was particularly effective over time, even in cases where the starting permeation values were significantly lower than those measured using pure water as the BF.

The most advantageous condition, suitable for combining the highest gas permeance and gas selectivity values, is represented by the H7 hollow fibers, obtained by feeding a salt solution as BF. In particular, for this sample the decrease in gas permeance was comparable to that observed for H2 and H6 membranes, while the selectivity for the different gas pairs was similar to the highest, measured for the crosslinked samples (H8 sample). As on the fresh samples, also upon aging, the results of the gas permeance of the H7 membranes were over three times higher than those of H8.

## 4. Conclusions

This study successfully combined a triple orifice spinneret with a bore fluid at a reduced water activity for preparing gas selective hollow fibers based on a polyimide. The HFs were prepared using solvent/water or salt/water mixtures as bore fluid. The morphological analysis evidences how the BF composition can modulate the skin thickness. Hansen and Gorgon parameters analysis helped to interpret the behavior of the system polymer-solvent-coagulant mixtures, justifying the morphology of the samples. The permeation properties of the hollow fibers are dependent on the water activity of the bore fluid and on the type of bore fluid.

Lowering the water activity in the bore fluid during the HF spinning reduces its inflow diffusion on the inner side and results in improved selectivity for gas separation. A high NMP concentration in the bore fluid (60 wt.%) results in enhanced viscosity and prevents solvent diffusion from the nascent HF. This condition results in thinner, but not extremely selective, skin layers. The incidence of these active layers in the permeation process was more important for the samples having a higher selectivity for the different gas pairs. Salt solutions are more effective in promoting the solvent outflow from the nascent than solvent solutions. Interesting gains in the selectivities for fast/slow gas pairs can be reached by adopting NaCl at 20 wt.% with respect to the same weight amount of solvent in the BF. This approach results in selectivity values that are extremely high, close to those obtained by adding a crosslinker agent in the BF, without compromising the gas permeance. Long-term gas permeation testing confirms the effectiveness of a salt-water BF solution in enhancing gas separation performances. The option of using a salt like NaCl in the bore fluid provides a greener preparation approach, reducing the amount of solvent required for the membrane preparation.

## Figures and Tables

**Figure 1 polymers-13-02211-f001:**
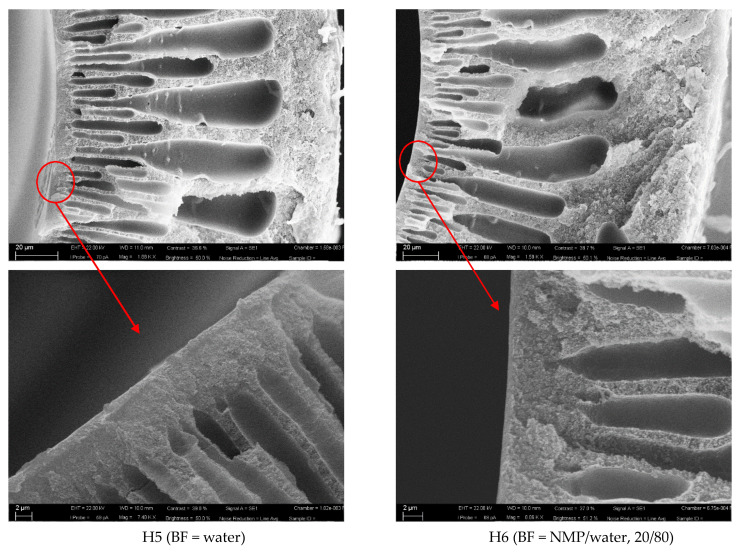
SEM images of H5 and H6 HFs: a cross-section and magnification of the inner skin layer.

**Figure 2 polymers-13-02211-f002:**
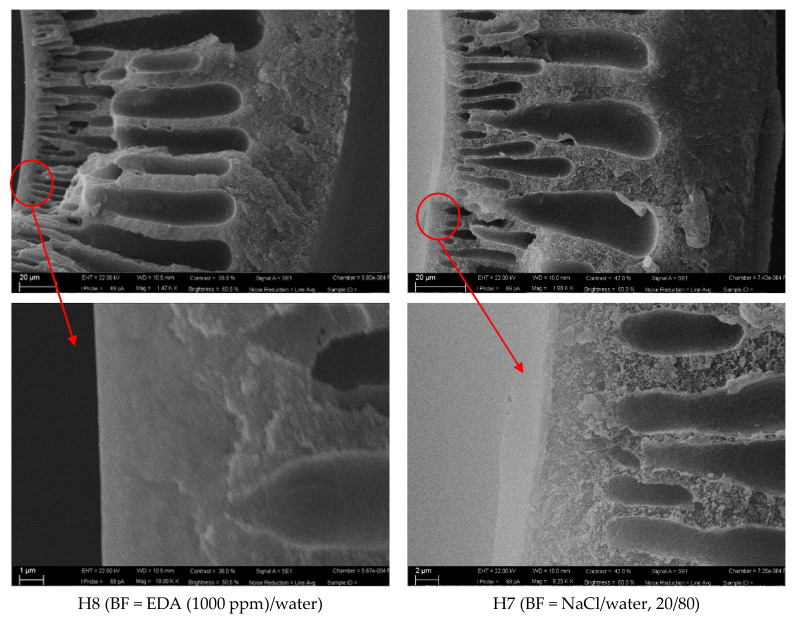
SEM images of H7 and H8 HFs: a cross-section and magnification of the inner skin layer.

**Table 1 polymers-13-02211-t001:** Operating conditions for the preparation of the polyimide HFs.

Dope Flow-Rate (g min^−1^)	3.6 or 5.0
**BF flow-rate** (g min^−1^)	3
**EF flow-rate** (g min^−1^)	3
**Air gap** (cm)	60
**T dope** (°C)	50
**T coagulation bath** (°C)	18

**Table 2 polymers-13-02211-t002:** Composition of the BF and EF used for the preparation of the HFs.

HF Code	Q_DOPE_ = g min^−1^	BF Composition (wt/wt)	EF Composition (wt/wt)
H0	5.0	water	-
H1	5.0	water	NMP/water (95/5)
H5	3.6		
H2	5.0	NMP/water (20/80)	NMP/water (95/5)
H6	3.6		
H3	5.0	NMP/water (30/70)	NMP/water (95/5)
H4	5.0	NMP/water (60/40)	NMP/water (95/5)
H7	3.6	NaCl/water (20/80)	NMP/water (95/5)
H8	3.6	EDA/water (1000 ppm)	NMP/water (95/5)

**Table 3 polymers-13-02211-t003:** Thickness of the inner top layer evaluated from SEM images of HFs prepared with different BF compositions.

HF Code	Thickness (nm)	BF Composition (wt/wt)	EF Composition (wt/wt)
H0	ca. 800	water	-
H1	ca. 800	water	NMP/water (95/5)
H2	ca. 700	NMP/water (20/80)	NMP/water (95/5)
H3	ca. 400	NMP/water (30/70)	NMP/water (95/5)
H4	ca. 250	NMP/water (60/40)	NMP/water (95/5)
H5	ca. 900	water	NMP/water (95/5)
H6	ca. 600	NMP/water (20/80)	NMP/water (95/5)
H7	ca. 700	NaCl/water (20/80)	NMP/water (95/5)
H8	ca. 3400	EDA/water	NMP/water (95/5)

**Table 4 polymers-13-02211-t004:** Permeation properties measured on the Matrimid^®^ HFs tested as prepared. Q_DOPE_ = 5.0 g min^−1^.

HF Code	BF	EF	Permeance (GPU)						Selectivity (-)		
			H_2_	He	CO_2_	O_2_	CH_4_	N_2_	O_2_/N_2_	CO_2_/N_2_	H_2_/N_2_
H0	water	-	42.3	36.3	13.9	3.52	0.50	0.62	5.68	22.4	68.2
H1	water	NMP/water (95/5)	46.4	39.7	21.5	4.03	0.54	0.645	6.25	33.3	71.9
H2	NMP/water (20/80)	NMP/water (95/5)	31.5	29.3	10.4	1.94	0.31	0.33	5.88	31.5	95.5
H3	NMP/water (30/70)	NMP/water (95/5)	45.7	38.9	21.0	3.83	0.59	0.70	5.47	30.0	65.3
H4	NMP/water (60/40)	NMP/water (95/5)	148	127	63.3	12.6	2.17	2.34	5.38	27.1	63.3
1 GPU = 10^−6^ cm^3^ cm^−2^ s^−1^ cmHg^−1^											

**Table 5 polymers-13-02211-t005:** Permeation properties measured on the Matrimid^®^ HFs tested as prepared. Q_DOPE_ = 3.6 g min^−1^.

HF Code	BF	EF	Permeance (GPU)						Selectivity (-)		
			H_2_	He	CO_2_	O_2_	CH_4_	N_2_	O_2_/N_2_	CO_2_/N_2_	H_2_/N_2_
H5	water	NMP/water (95/5)	41.1	36.5	18.2	3.53	0.51	0.58	6.10	31.4	71.4
H6	NMP/water (20/80)	NMP/water (95/5)	34.4	31.4	10.6	2.0	0.284	0.353	5.67	30.0	97.5
H7	NaCl/water (20/80)	NMP/water (95/5)	47.2	42.1	17.1	3.2	0.44	0.485	6.49	34.7	97.3
H8	EDA/water 1000 ppm	NMP/water (95/5)	17.0	16.1	5.2	1.05	0.12	0.156	6.73	33.3	109
1 GPU = 10^−6^ cm^3^ cm^−2^ s^−1^ cmHg^−1^											

**Table 7 polymers-13-02211-t007:** Permeation properties measured on the HFs as prepared and after 18 months.

HF Code	BF	EF	Testing	Permeance (GPU)						Selectivity (-)		
				H_2_	He	CO_2_	O_2_	CH_4_	N_2_	O_2_/N_2_	CO_2_/N_2_	H_2_/N_2_
H0	water	-	As-spun	42.3	36.3	13.9	3.52	0.50	0.62	5.68	22.4	68.2
			Aged	20.0	19.3	4.94	1.24	0.16	0.21	5.90	23.5	95.2
H1	water	NMP/water (95/5)	As-spun	46.4	39.7	21.5	4.03	0.54	0.645	6.25	33.3	71.9
			Aged	23.9	23.6	8.03	1.68	0.212	0.25	6.72	32.1	95.6
H2	NMP/water (20/80)	NMP/water (95/5)	As-spun	31.5	29.3	10.4	1.94	0.31	0.33	5.88	31.5	95.5
			Aged	24.4	24.3	8.51	1.7	0.199	0.254	6.69	33.5	96.1
H5	water	NMP/water (95/5)	As-spun	41.1	36.5	18.2	3.53	0.51	0.58	6.10	31.4	71.4
			Aged	21.6	21.3	7.84	1.56	0.198	0.24	6.50	32.7	90.0
H6	NMP/water (20/80)	NMP/water (95/5)	As-spun	34.4	31.4	10.6	2.0	0.284	0.353	5.67	30.0	97.5
			Aged	23.6	22.9	7.37	1.51	0.183	0.242	6.24	30.5	97.5
H7	NaCl/water (20/80)	NMP/water (95/5)	As-spun	47.2	42.1	17.1	3.2	0.44	0.485	6.49	34.7	97.3
			Aged	31.5	30.7	9.22	1.88	0.252	0.27	6.96	34.1	117
H8	EDA/water (1000 ppm)	NMP/water (95/5)	As-spun	17	16.1	5.2	1.05	0.12	0.156	6.73	33.3	109
			Aged	10.6	10.0	2.83	0.586	0.071	0.085	6.89	33.3	125

## Data Availability

The data presented in this study are available from the corresponding author upon reasonable request.

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
