# Peer review of "Hollow Fiber Polyimide Membranes Prepared in a Triple Orifice Spinneret: Effect of a Reduced Water Activity in the Bore Fluid on the Gas Separation Performance"

_polymers, 2021, doi:10.3390/polym13132211_

Round 1
Reviewer 1 Report
In this paper, the authors have prepared a set of hollow fiber membranes by modulating bore fluid and dope solution. This topic is of course important and helps to design the membranes for various applications.
However, I have concerns over the novelty. Playing with the solution conditions and the bore has been studied for a long time. Some of the cited references even say this. Therefore, I would recommend adding more details before being published.
1. The morphology (as seen by SEM) has changed a bit and the selective layer thickness is changing resulting in a slight variation in the gas permeances. Add SEM images for the remaining samples (in Supplementary).
2. As the core focus of the manuscript is on the solution and bore fluid conditions. I would strongly recommend presenting the kinetics data resulting from interaction with the coagulant and the bore fluids.
3. It is also recommended to add the active layer thickness of all the membranes for which permeation data is given in Table 4.
4. When we calculate the permeability (thickness normalized permeance) it is not the same. This means the active layer is not the major part defining gas permeance. More details should be added concerning the interaction between the solvents.
5. Since a mixture of solvents is being used, Hansen solubility parameters can be a good estimate in defining the final morphology.
Author Response
We would like to thank the Reviewer for the kind evaluation of the manuscript that helped us to improve its quality.

Reviewer 2 Report
The manuscript P. Bernardo et al "Hollow fiber polyimide membranes prepared in a triple orifice spinneret: effect of a reduced water activity in the bore fluid on the gas separation performance" is devoted to the production of hollow fiber membranes. The authors have correctly structured the work as follows, they discuss issues not only devoted to the formation of morphology depending on the type of precipitator, but also the rheological properties of systems.
In my opinion, the work can be published in the journal, but it is desirable to add information on the mechanical properties of the samples obtained.
Author Response
We thank the Reviewer for the consideration of our work and we appreciate the useful suggestions.

Round 2
Reviewer 1 Report
I agree with the revised version and support it’s publication.